# HSV1716 Prevents Myeloma Cell Regrowth When Combined with Bortezomib *In Vitro* and Significantly Reduces Systemic Tumor Growth in Mouse Models

**DOI:** 10.3390/v15030603

**Published:** 2023-02-22

**Authors:** Simon Tazzyman, Georgia R. Stewart, James Yeomans, Adam Linford, Darren Lath, Joe Conner, Munitta Muthana, Andrew D. Chantry, Michelle A. Lawson

**Affiliations:** 1Sheffield Myeloma Research Team, University of Sheffield Medical School, University of Sheffield, Beech Hill Road, Sheffield S10 2RX, UK; 2Mellanby Centre for Musculoskeletal Research, Medical School, University of Sheffield, Beech Hill Road, Sheffield S10 2RX, UK; 3Department of Oncology and Metabolism, University of Sheffield Medical School, University of Sheffield, Beech Hill Road, Sheffield S10 2RX, UK; 4Sorrento Therapeutics, 4955 Directors Place, San Diego, CA 92121, USA

**Keywords:** oncolytic virus, HSV1716, herpes simplex virus, multiple myeloma, apoptosis, systemic murine models

## Abstract

Multiple myeloma remains largely incurable due to refractory disease; therefore, novel treatment strategies that are safe and well-tolerated are required. Here, we studied the modified herpes simplex virus HSV1716 (SEPREHVIR^®^), which only replicates in transformed cells. Myeloma cell lines and primary patient cells were infected with HSV1716 and assessed for cell death using propidium iodide (PI) and Annexin-V staining and markers of apoptosis and autophagy by qPCR. Myeloma cell death was associated with dual PI and Annexin-V positivity and increased expression of apoptotic genes, including CASP1, CASP8, CASP9, BAX, BID, and FASL. The combination of HSV1716 and bortezomib treatments prevented myeloma cell regrowth for up to 25 days compared to only transient cell growth suppression with bortezomib treatment. The viral efficacy was tested in a xenograft (JJN-3 cells in NSG mice) and syngeneic (murine 5TGM1 cells in C57BL/KaLwRijHsd mice) systemic models of myeloma. After 6 or 7 days, the post-tumor implantation mice were treated intravenously with the vehicle or HSV1716 (1 × 10^7^ plaque forming units/1 or 2 times per week). Both murine models treated with HSV1716 had significantly lower tumor burden rates compared to the controls. In conclusion, HSV1716 has potent anti-myeloma effects and may represent a novel therapy for multiple myeloma.

## 1. Introduction

Multiple myeloma (MM) is a hematological malignancy characterized by the clonal expansion of terminally differentiated B lymphocytes, known as plasma cells, in the bone marrow (BM). It is the second most common hematological malignancy, with an incidence rate that has increased by 66% since the 1970s [1]. Despite substantial improvements being made in overall survival with the introduction of more effective anti-myeloma agents, the disease usually relapses and becomes refractory to treatment, resulting in early death. Therefore, strategies to eliminate residual disease and prevent relapse after chemotherapy are needed. 

Recent advances in oncolytic virus (OV) therapy have shown promise in several cancers preclinically and clinically, including myeloma [2,3,4,5,6], with FDA approval of talimogene laherparapvec (T-Vec), a modified herpes simplex virus-1 (HSV-1), which encodes granulocyte–macrophage colony-stimulating factor (GM-CSF), for the treatment of advanced melanoma [7,8]. OVs preferentially infect and kill malignant cells using multiple mechanisms to eradicate tumor cells, including promoting anti-tumor immune responses [4,9], although OVs are typically modified to enable tumor specificity [10,11].

HSV1716 (SEPREHVIR) is a modified HSV-1 with deletions in both copies of the RL1 gene, encoding the virulence factor ICP34.5. ICP34.5 binds to protein phosphatase 1α, enabling protein synthesis [12], and proliferating cell nuclear antigen (PCNA), switching it from repair to replication [13]; without ICP34.5, HSV1716 cannot replicate [14]. These mechanisms are disturbed in the tumor setting, ensuring tumor specificity; PCNA is activated in tumors and HSV1716 susceptibility is linked to its expression in cells [15]. Additionally, oncogenes such as RAS and MEK inhibit the PKR pathway, negating the virally induced shutdown of protein synthesis [16]. Preclinical models with HSV1716 in glioma showed a reduced tumor burden and viral persistence, with replication seen only in tumor tissues [17,18,19,20]. Similar results have been observed in melanoma [21,22]. Recent investigations show that oncolytic HSVs can infect myeloma cells, induce apoptosis, induce oncolysis in cell lines and primary patient samples, and reduce tumor burden in subcutaneous xenograft models of myeloma [23,24].

Here, we show using *in vitro* assays that HSV1716 is effective at inducing oncolysis in both myeloma cell lines and *ex vivo* primary patient-derived myeloma cells, and when HSV1716 is combined with bortezomib it prevents myeloma cell regrowth. We also show for the first time that HSV1716 significantly reduces tumor burden in systemic xenograft and syngeneic murine models of myeloma. 

## 2. Materials and Methods

### 2.1. Tissue Culture

Human JJN-3, OPM-2 (DSMZ, Germany), U266 (LGC Standards, London, UK), RPMI-8226 (ATCC^®^ (CCL-155™), Manassas, VA, USA), and murine 5TGM1 (Dr Oyajobi, University of Texas, San Antonio, TX, USA) myeloma cell lines were maintained in complete RPMI medium as previously described [25]. The cell lines were genetically profiled by DSMZ and ATCC using a short tandem repeat (STR) analysis to confirm their identity and were routinely tested for mycoplasma.

Patient samples were acquired under research ethics number 05/Q2305/96 and informed consent was obtained from all subjects involved in the study. Human primary lymphocytes were isolated from waste buffy coats using Ficoll separation as described previously [26]. The peripheral blood mononuclear cell (PBMC) layer was collected and seeded at 7 × 10^7^ cells/flask overnight in IMDM, 2% human AB serum, 1% P/S. Unattached lymphocytes were collected and used in infection experiments. BM aspirates were obtained from myeloma and plasma cell leukemia (PCL) patients and from healthy donors. Primary plasma cells were isolated using magnetic activated cell sorting with CD138^+^ Microbeads (Miltenyi Biotech, Woking, UK). The CD138^+^ and CD138^-^ BM fractions were cultured in complete RPMI medium [25] and DMEM with 2 mM L-glutamine, 10% FCS, and 1%P/S (100 U/100 μg/mL), respectively. Both cell populations were used in infection experiments.

All tissue culture reagents were from Thermo Fisher Scientific (Runcorn, UK) and all cells were cultured at 37 °C in 5% CO_2_ unless stated otherwise.

### 2.2. HSV1716-GFP Infection Assays

Here, 2 × 10^5^ JJN-3 and U266 myeloma cell lines were infected with HSV1716-GFP (SEPREHVIR^®^, provided by Sorrento Therapeutics, San Diego, CA, USA [18]). HSV1716 has GFP inserted into the RL1 gene locus driven by the phosphoglycerate kinase promoter. At 24, 48, and 72 h post-infection, the cells were stained with TO-PRO-3 and the GFP was assessed on viable TO-PRO-3 negative cells using a BD LSRII flow cytometer (BD Biosciences, Wokingham, UK). 

### 2.3. Oncolysis Assays

Cell lines seeded at 10^5^ cells/well in 1 mL of medium were exposed to HSV1716 in the range of multiplicity of infection (MOI). Cell death was assessed by propidium iodide (PI) staining (2 µg/mL) analyzed on an Attune™ Flow Cytometer (Thermo Fischer Scientific, Runcorn, UK).

The combination treatment (bortezomib with HSV1716) was assessed by treating 10^5^ cells/well with the vehicle (PBS) or 2.5 nM of bortezomib for 1 day. After 2 and 4 days, the viability was assessed by PI staining using flow cytometry. At day 4, the bortezomib-treated cells were divided into two groups, with half treated with PBS and the other half with HSV1716 (MOI of 5). Their viability was determined after 11, 18, and 25 days.

For primary myeloma patient and healthy donor samples, CD138^+^ cells and CD138^-^ BM or PBMC cells were seeded at 2 × 10^5^ cells/mL of media. Both cell populations were treated with HSV1716 MOI 5 or the vehicle (PBS) control in the presence of 10% autologous serum. After 96 h, cell death was assessed by PI staining using flow cytometry on a BD FacsCalibur system (Becton Dickinson, Oxford, UK). For primary lymphocyte assays, the lymphocyte population was gated using standard FSc and SSc properties and the cell viability and counts were assessed as above.

### 2.4. ICP0 and ICP8 Gene Expression Analyses

Here, 10^6^ JJN-3 cells were infected with control or HSV1716 MOI 5 for 8, 16, and 24 h. The total RNA was isolated from cells using a ReliaPrep RNA cell MiniPrep system (Promega, Southampton, UK) and the cDNA was synthesized using a High-Capacity RNA-to-cDNA kit (Thermo Fisher Scientific). The qPCR was performed using primers specifically targeting the viral genes ICP0 and ICP8 (Appendix A) with GAPDH as a housekeeping gene, then assessed using SYBR Green (Primer Design, Southampton, UK) with 2× SYBR^®^ Green Select Master Mix (Thermo Fisher Scientific) and detected with an ABI Prism 7900HT system and SDS 2.1 software (Applied Biosystems, Foster City, CA, USA).

### 2.5. Apoptosis Analysis

Here, 10^6^ JJN-3 cells were infected with the control or HSV1716 MOI 5 for 8–24 h. The total RNA was isolated, and the cDNA was synthesized as described in Section 2.4. The gene expression levels of FASL (Hs0018225_m1) and BCL2 (Hs04986394_s1) were detected using Taqman assays^®^ (Life technologies, Glasgow, UK). Additionally, the cells were infected for 24 h and stained with Annexin-V-APC (Biolegend, London, UK) and PI.

### 2.6. PrimePCR^TM^ Human Apoptosis Microarray Array 

Here, 10^6^ JJN-3 cells were infected with the control or HSV1716 MOI 5 for 24 h. The total RNA was isolated, and the cDNA was synthesized as described in Section 2.4. A PrimePCR^TM^ SYBR^®^ Green-based microarray (BIO-RAD, Watford, UK) was used to detect the expression of multiple apoptotic genes using RT-qPCR as per the manufacturer’s instructions.

### 2.7. JJN-3 Xenograft Model of Myeloma

The animal procedures were governed by the University of Sheffield UK Home Office License 70/8670 in accordance with the Animal Act 1986. The group sizes were calculated from previous studies [27] with a desired power of 80% and alpha level of 0.05 using a mean of 48.54 and standard deviation of 7.48, with a desired reduction rate of 30% of the tumor burden, giving a sample size of 5.

Ten female 6–7 week-old NSG mice (*n* = 5/group) received 100 μL of 10^6^ JJN-3 cells by intravenous (i.v.) injection. After 6 days the mice were randomized into 2 groups and treated with 100 μL of PBS or 10^7^ plaque forming units (P.F.U) of HSV1716 by i.v. injection, and treatments were repeated on days 12 and 18 post-tumor inoculation.

Left tibiae and femora, right tibiae, livers, and spleens were fixed in buffered formalin and embedded in paraffin wax. The BM cells of the right femora were flushed with 500 μL of PBS and the tumor burden was analyzed using flow cytometry analysis using FITC-labeled anti-human HLA [25].

### 2.8. 5TGM1 Syngeneic Murine Model of Myeloma

Ten female 6–7-week-old C57BL/6KalWRij mice (*n* = 5/group) received 100 μL of 10^6^ 5TGM1 cells by i.v. injection. One week later, the mice were randomized into 2 groups and treated with 100 μL of PBS or HSV1716 (10^7^ P.F.U) twice weekly by i.v. injection. The mice were culled after 21 days, and the organs were collected. The tumor burden was assessed via the cell morphology of hematoxylin and eosin (H&E)-stained sections of tibiae.

### 2.9. Assessment of Tumor Burden

The tibiae from JJN-3 xenograft mice were formalin-fixed, decalcified, and paraffin-embedded. Then, 3 µm sections were stained with anti-HSV antibody (Sorrento Therapeutics, San Diego, CA, USA), then slides were scanned on a Hamamatsu NanoZoomer XR (Hamamatsu, Hertfordshire, UK) and viral infection was assessed in ImageScope (Lieca Biosystems, Newcastle, UK).

Formalin-fixed, decalcified, and paraffin-embedded tibial sections from 5TGM1 syngeneic mice were H&E-stained to assess them for tumor burden, then slides were scanned on a Hamamatsu NanoZoomer XR and viral infection was assessed in ImageScope.

### 2.10. Assessment of Bone Disease

Tibiae were fixed in 10% formalin and scanned using a SkyScan 1272 system (Bruker, Kontich, Belgium), as previously described [28]. All measurements followed standard guidelines [29]. Next, 3 µm decalcified wax tissue sections were assessed for osteoclasts and osteoblasts as previously described [30]. 

### 2.11. Statistical Analyses

The data were assessed for normality using the D’Agostino–Pearson test and relevant parametric or non-parametric statistical tests. Where a normality test could not be performed, normality was assumed and either a student’s T-test or ANOVA was applied with a Bonferroni post-test.

## 3. Results

### 3.1. HSV1716 Induces Potent Cell Death in Human Myeloma Cell Lines and Primary Patient Samples

JJN-3 and U266 cells were infected with HSV1716-GFP at MOIs of 0.5 and 5. After 24, 48, and 72 h, the GFP expression was assessed by flow cytometry (Figure 1(ai–aiii)). The JJN-3 and U266 cells showed significant GFP expression as early as 24 h, indicating myeloma cell line susceptibility to HSV1716 infection. The JJN-3 cells were infected with increasing concentrations of HSV1716 at MOIs of 0.5–100. After 4, 5, and 6 days of infection, the cell numbers were significantly lower than in the controls (Figure 1(bi)). The HSV1716 (MOI 5) infection resulted in significantly lower cell numbers in JJN-3 and U266 myeloma cell lines at 48 and 72 h compared to the controls (Figure 1(bii)). HSV1716 (MOI 5) reduced the cell viability in JJN-3, RPMI-8226, U266, and OPM-2 cells as indicated by an increase in PI-positive cells after 4 days compared to the control (Figure 1c). To confirm the HSV1716 gene expression in myeloma cells, we investigated the production of the viral transactivator gene ICP0 and replication gene ICP8. As expected, the HSV1716 infection increased the expression of both genes after 8 and 24 h when compared to the control cells, demonstrating viral transcription (Figure 1(di,dii)).

To ensure HSV1716 was not toxic to healthy cells, primary lymphocytes and BM samples (CD138^−^ and CD138^+^ fractions) from healthy donors were infected with HSV1716 (MOI of 0.5 or 5) and the cell viability was assessed at 4, 5, and 6 days post-infection. No difference in PI positivity was seen between the control cells and the HSV1716-infected cells (Figure 1(ei,eii)). Next, we investigated the effects of HSV1716 on primary patient myeloma or plasma cell leukemia (PCL) cells. The cells were treated with HSV1716 (MOI 5) and after 4 days the viability was assessed using PI. The treatment with HSV1716 (Figure 1f) on CD138^+^ cells caused significant increases in cell death when compared to the control-treated groups. There was considerable variation in the percentages of PI positivity in the control-treated cells. For example, in the myeloma cells, these ranged from a low of 8.7% to a high of 58.9%. However, no matter what the basal level of cell death, this was substantially increased following viral therapy. The average fold change for myeloma patients between control and HSV1716-treated cells was 2.7. 

To determine if cells recovered after long-term infection, the JJN-3 cells were infected with HSV1716 at an MOI of 5 and the cell counts were assessed at 4, 11, 18, and 25 days post-infection. The virally infected cells had lower cell numbers compared to the control cells at every time point but were significantly decreased in number between days 18 and 25 (Figure 1g). However, if used in a clinical setting, HSV1716 is unlikely to be given as a monotherapy. Instead, HSV1716 would be given alongside such standard therapies as bortezomib. Therefore, the JJN-3 cells were treated with bortezomib, resulting in significant increases in PI staining after 2 and 4 days. The cells treated only with bortezomib recovered, with the viability returning to levels seen in control-treated cells. When bortezomib was combined with HSV1716, cell regrowth was prevented, and the cells remained close to 90% PI-positive between 11 and 25 days (Figure 1h).

### 3.2. HSV1716 Induces Cell Death via Apoptosis

To determine how oncolytic viruses induce cell death, infected myeloma cells were analyzed by qPCR and flow cytometry. HSV1716 induced a significant increase in the number of dual-stained Annexin-V- and PI-positive necrotic cells (Figure 2(ai,aii)). This was matched by a significant reduction in the number of healthy cells. To further assess apoptosis, the FASL and BCL2 expression was measured. Following the HSV1716 infection, FASL (Figure 2b) showed a 6601-fold increase at 24 h. No difference was seen for the anti-apoptotic BCL2. To further assess apoptosis, a cell death microarray was performed, which showed fold increases in 25 pro-apoptotic genes, including CASP1, CASP8, CASP9, BAX, and BID (Figure 2c). Therefore, HSV1716 may utilize the apoptotic pathway to induce myeloma cell death.

### 3.3. Oncolytic Virus Therapy Significantly Lowers Tumor Burden in Murine Myeloma Models

The HSV1716 infection reduced the myeloma viability *in vitro* and had little impact on the control cells. We, therefore, wanted to determine the impact of the HSV1716 treatment in preclinical murine models. To test the efficacy of HSV1716 we used 2 *in vivo* models of myeloma (xenograft JJN-3 and the syngeneic 5TGM1) (Figure 3a,e). In both, the mice were treated systemically with HSV1716 at 1 × 10^7^ P.F.U.

At day 21, mice bearing xenograft JJN-3 tumors were sacrificed and BM was flushed from mouse femurs. The tumor burden was assessed by HLA staining and flow cytometry; the JJN-3 control mice had a tumor burden of 41.78 ± 0.97%, while the HSV1716-treated mice had a significantly lower tumor burden of 20.73 ± 5.11% (Figure 3(bi,bii)). The BM sections of tibiae were assessed for HSV viral particles by IHC. The tibiae from HSV1716-treated animals were positive for HSV1716, while the PBS-treated animals were negative (Figure 3(ci,cii) and Appendix A). To determine the location of viral infection, higher power images of HSV1716-treated tibiae were assessed. The analysis demonstrated that HSV viral particles were only detected in regions of the BM made up of JJN-3 myeloma cells (Figure 3(di,dii) and Appendix A). Because myeloma induces bone disease, we investigated whether the HSV1716 treatment altered the trabecular bone parameters due to a reduced tumor burden. The HSV1716-treated mice had a significantly higher trabecular volume compared to PBS-treated mice; there was a trend for an increase in trabecular thickness, but this was less pronounced and did not reach significance (Appendix A). Because of this increase in trabecular bone volume, we investigated whether there were changes to bone remodeling. Histomorphometry measurements revealed that there was a trend for increased osteoblasts and decreased osteoclasts, but this did not reach significance (Appendix A).

We then tested the efficacy of HSV1716 in the 5TGM1 murine syngeneic model of myeloma [31]. Prior to tumor inoculation, HSV1716 was tested *in vitro* on the 5TGM1 cells. The cells were infected and the viability was assessed after 4 and 7 days (Appendix A). In contrast to human myeloma cell lines, there was no difference in PI positivity after 4 days. However, after 7 days, the PI positivity in the cells treated at an MOI of 5 was significantly higher than in the controls. *In vivo* 5TGM1-bearing mice were treated systemically with HSV1716 or PBS and their tumor burden was assessed histologically following H&E staining in both tibiae. The virally treated mice showed significant reductions compared to the vehicle-control-treated mice of 28.36 ± 7.46% and 77.94 ± 5.43%, respectively (Figure 3(fi,fii,g)). We then assessed myeloma bone disease via micro-CT and found there was a trend for an increased trabecular bone volume. However, this was less pronounced than in the xenograft model and did not reach significance; similar results were obtained for the trabecular number (Appendix A).

## 4. Discussion

Several previous publications have demonstrated the efficacy of HSV1716 in other cancers [18,26,32,33,34,35,36], with the data showing that modified HSVs have preclinical efficacy in myeloma [23,24]. Our data presented here support the previous publications with important additional details. Our *in vitro* data demonstrate a rapid impact of HSV1716 on four human myeloma cell lines. Significant increases in cell death of 50–80% were achieved when compared to untreated controls. This range of cell death observed here was similar to that seen in other cells *in vitro* (glioma [19] and head and neck cancer [33] for HSV1716) and in myeloma cell lines *in vitro* [24]. We also saw an induction of the viral transactivator and replication genes ICP0 and ICP8, respectively [37]. Gene activation was observed at 8 and 24 h post-HSV1716 infection, but not at 16 h, for both ICP0 and ICP8; at 24 h, this likely represents either a second round of infection or the detection of the viral genome following a potential first round of transactivation at 8 h. This rapid transactivation of the viral genome and tumor lysis was consistent with what has been seen previously with viral particle production and release in the first few days following infection [20,21,38]. Interestingly, Ghose et al. [24] found a modified HSV to not replicate in human myeloma cell lines, as assessed by plaque-forming assays, despite causing cytotoxicity in myeloma cells. However, they found that a replication-competent virus rather than heat-inactivated HSV virus was needed to induce cell cytotoxicity; therefore, they concluded that the HSV-induced cytotoxicity is independent of HSV replication in myeloma cells. 

The data presented here demonstrate that HSV1716 is specific for myeloma tumor cells, with minimal or a lack of lysis observed in healthy BM cells flushed from patients. Importantly, it is often noted that viral therapy is effective against tumor cell lines *in vitro* and *in vivo*, with little impact in a clinical setting. In contrast, we have shown that HSV1716 effectively reduces tumor viability in primary isolated myeloma cells. This is exciting, as the heterogeneity of myeloma tumors presents a potential barrier to these therapies that may not be evident when using cell lines. The heterogeneity of myeloma is well known, with the continual emergence of refractory clones, meaning that viral activity is not guaranteed. Additionally, primary cells tend to proliferate more slowly than cell lines [39,40], which was of particular concern for HSV1716. Ghose et al. published similar findings with a modified HSV and saw a reduction in primary patient-derived myeloma cells. However, cell death was assessed at earlier time points (24 and 48 h), and this was likely to be the reason why they did not observe as high of an effect at the same MOI as what is presented here [24]. 

Myeloma is a heterogenous malignancy, with frequent disease relapse and chemoresistance [41] seen in patients. Interestingly, we observed a consistent decrease in cell number and no recovery in tumor cell regrowth for up to 25 days post-infection in JJN-3 cells following treatment with HSV1716 combined with standard-of-care treatment with bortezomib. In other studies, the clonogenic recovery of chemotherapy-treated myeloma cells was observed at 14 to 21 days post-treatment [42]. It is, therefore, possible that long-term treatment with HSV1716 will be successful as a vital and unique component of combined therapy to prevent myeloma regrowth, as HSV1716 kept cell numbers down beyond this time frame. The recovery of tumor cells in a clinical setting is likely to be modulated by the immune system. Activating the immune system will theoretically prevent tumor cell regrowth. Oku et al. found an increased anti-myeloma effect using a modified HSV when myeloma cells were cultured with PBMCs through activating plasmacytoid dendritic cells and natural killer cells. This effect was enhanced when the cells were treated in combination with lenalidomide [23]. Our data combined with the data published by Oku and colleagues suggested the possibility that HSV1716 will be successful at preventing myeloma regrowth alone or as a combined therapy.

The method by which HSV1716 virus induces tumor cell death appears to be via apoptosis. This is in agreement with other studies where HSV-1 has been seen to induce cell death through apoptosis or direct bursting due to viral build-up [43]. Our study demonstrates that cell death leads to the induction of pro-apoptotic genes such as FASL and CASP1, -8, and -9. Previous publications suggest that HSV-induced apoptotic cell death happens quickly, while other mechanisms of cell death are more prolonged. For example, cell death via direct bursting requires 60–70 h before death is observed [43]. Considering these observations and the rapid induction of cell death (just 24–48 h), it is likely that in myeloma HSV1716 induces cell death via apoptosis. These data are in support of the data published by Ghose et al. [24], who demonstrated that myeloma cells lines show increased Annexin-V positivity and increased protein levels of cleaved caspase 3 and cleaved PARP in response to a modified form of HSV. The primary patient-derived myeloma cells also appeared to have increased Annexin-V positivity following modified HSV infection. 

The efficacy of viral therapy *in vitro* was matched by similar efficacy in murine myeloma models. The systemic injection of HSV1716 resulted in a significant 50% reduction in myeloma xenografts. A similar efficacy was also seen in the immune-competent 5TGM1 model, which resulted in a significant 63.6% reduction in tumor burden. These data are promising, as they show the modified HSV to have efficacy in a systemic model of myeloma, with the virus being administered systemically. Unfortunately, it is difficult to compare our findings to previous publications using modified HSVs or other OVs, as the majority have only used subcutaneous myeloma models with intratumoral injections, which do not have clinical relevance in many myeloma patients, as the systemic cancer forms primarily in the BM. However, there are some publications reporting on the efficacy of OVs in systemic murine models of myeloma with similar efficacy to what we observed, such as reovirus, which showed a 50% reduction in tumor burden compared to the control in the 5TGM1 systemic syngeneic model [9].

The reduction in tumor burden observed in the xenograft model had pronounced effects on preventing myeloma-induced bone disease *in vivo*, resulting in the significant inhibition of the tumor-induced loss of trabecular bone volume. This was probably as a direct effect of the decreased tumor volume in HSV1716-treated mice. In the syngeneic model, this inhibition of the tumor-induced loss of trabecular volume was less pronounced and not significant. Histologically, xenograft mice treated with HSV1716 have shown trends for decreased osteoclast numbers and increased osteoblast numbers, although not significantly. 

There are emerging studies showing the efficacy of the systemic delivery of viruses in numerous tumor settings, including breast, prostate, and hepatocellular models [11,34,44,45]. In our hands, the anti-myeloma response seen toward HSV1716 therapy is substantial and shows similar efficacy to other studies. Importantly, in our *in vitro* experiments, we observed that HSV1716 not only maintained the long-term inhibition of myeloma cell viability but also prevented the recovery of cell viability once the standard therapy (bortezomib) was removed. These data provide evidence for future murine studies to test combination therapies. 

In summary, we have shown that HSV1716 induces oncolysis in myeloma-patient-derived cells, prevents myeloma cell line regrowth when combined with bortezomib, and in systemic murine models HSV1716 alone reduces the myeloma burden. This provides further evidence that HSV1716 could be developed for use in myeloma. If so, it is likely to be combined with conventional myeloma treatments as a method for reducing minimal residual disease. We are now investigating methods for increasing the viral delivery to myeloma cells in the BM using strategies such as cellular delivery [27] or molecular targeting [34], and combining the anti-myeloma treatment with HSV1716 to ascertain whether the elimination of residual myeloma is possible after induction chemotherapy.

## Figures and Tables

**Figure 1 viruses-15-00603-f001:**
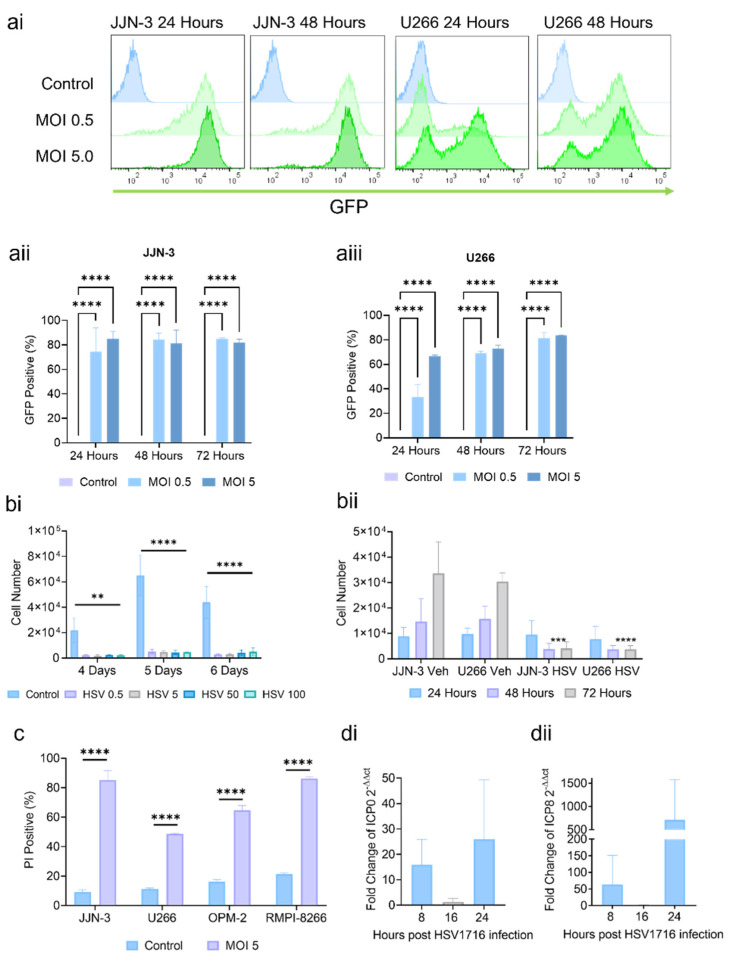
HSV1716 induces potent cell death in myeloma cells. (**a**) JJN-3 and U266 cells were infected with HSV1716-GFP at an MOI of 0.5 or 5. After 24, 48, and 72 h of infection, GFP expression was assessed using flow cytometry on viable cells. (**ai**) Representative flow cytometry GFP histogram plots of JJN-3 or U266 cells at 24 or 48 h after infection with the control, HSV1716 MOI 0.5, or HSV1716 MOI 5. GFP expression was plotted in (**aii**) JJN-3 and (**aiii**) U266 cells. Data are means ±SEMs, n = 3, **** = *p* < 0.0001 with respect to the control, analyzed using a two-way ANOVA. (**b**) JJN-3 or U266 cells were infected with HSV1716 at increasing MOIs (5–100). (**bi**) The cell count analysis shows significant reductions in cell numbers 4 days post-infection. Data are means ±SEMs, n = 3, **= *p* < 0.01, **** = *p* < 0.0001 with respect to the control, analyzed using a two-way ANOVA. (**bii**) Analyses at 24, 48, and 72 h showed a rapid response to HSV1716 in both myeloma cell lines. Data are means ±SEMs, n = 3, *** = *p* < 0.001, **** = *p* < 0.0001 with respect to the control, analyzed using a one-way ANOVA. (**c**) Four myeloma cell lines were infected with HSV1716 for 4 days and the viability was assessed using a flow cytometry analysis of PI-stained cells. All myeloma cell lines showed a significant increase in PI positivity. Data are means ±SEMs, n = 3, **** = *p* < 0.0001 with respect to the control, analyzed using a one-way ANOVA. (**d**) JJN-3 cells were infected with HSV1716 at an MOI of 5. At 8, 16, and 24 h post-infection, the total RNA was extracted and an RT-qPCR assay was performed to detect viral transactivator and replication genes ICP0 (**di**) and ICP8 (**dii**), both of which were significantly upregulated at both 8 and 24 h. (**ei**) Primary human lymphocytes from healthy donors were infected with HSV1716 at MOIs of 0.5 and 5. Cell viability was assessed 4, 5, and 6 days post-infection using a flow cytometry analysis and showed no significant increase in PI staining at any time point. Data are means ±SEMs, n = 3, analyzed using a one-way ANOVA. (**eii**) CD138^+^ and CD138^−^ cells isolated from BM aspirates from healthy donors were treated with HSV1716 at MOI 5 and the cell viability was assessed 4 days post-infection using a flow cytometry analysis and PI staining. Data are means, n = 3, analyzed using a Student’s T-test. (**f**) CD138^+^ cells isolated from BM aspirates from myeloma (MM) or plasma cell leukemia patients (PCL) who were treated with HSV1716 at an MOI of 5. Cell viability was assessed 4 days post-infection using a flow cytometry analysis and PI staining. HSV1716 induced a significant increase in cell death in primary CD138^+^ MM and PCL cells compared to untreated controls. Data are means, n = 6 for MM, n = 3 for PCL, * = *p* < 0.05 with respect to the control, analyzed using an ANOVA with Sidak’s multiple comparison. (**g**) Infection of myeloma cells lines with HSV1716 results in a long-term reduction in the myeloma cell number. Infected, but not control, JJN-3 cells showed consistent decreases in number between 4 and 25 days post-infection. Data are means ±SEMs, n = 3, **** = *p* < 0.0001, with respect to the control, analyzed using a two-way ANOVA. (**h**) JJN-3 cells were treated with PBS (Control) or 2.5 nM of bortezomib at day 0, after one day the cells were washed to remove bortezomib, and the viability was assessed at day 2 and day 4. Bortezomib groups were then split into two and treated with PBS (bortezomib only) or HSV1716 at MOI 5 (bortezomib + HSV1716). Data are means ±SEMs, n = 3, *** *p* = <0.001, **** *p* = <0.0001, # denotes significance between bortezomib HSV176 treatment and bortezomib, * denotes significance between bortezomib HSV1716 treatment and control, analyzed using a two-way ANOVA.

**Figure 2 viruses-15-00603-f002:**
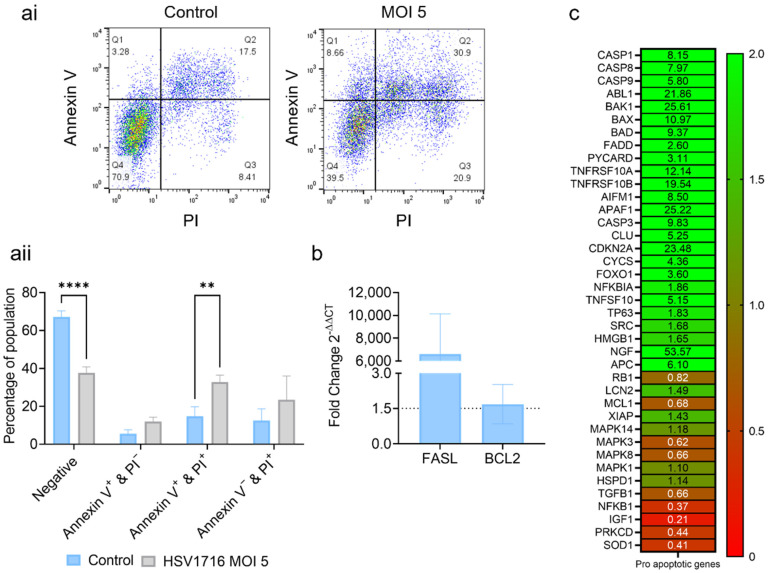
HSV1716 induces tumor apoptosis. JJN-3 cells were infected with HSV1716 at an MOI of 5 for 24 h and a flow cytometry analysis of Annexin-V and PI staining was carried out to assess apoptosis. (**ai**) Flow cytometry scatter plots showing Annexin-V vs. PI staining in control or HSV1716-treated cells. (**aii**) Percentages of cells in each population after Annexin-V and PI staining. Data are means ± SEMs, n = 3, ** = *p* < 0.01, **** = *p* < 0.0001 with respect to the relevant control, analyzed using a T-test. (**b**) The qPCR analysis of JJN-3 cells 24 h post-infection as fold changes over untreated controls. The analysis for the pro-apoptotic gene FASL and anti-apoptotic gene BCL2 following HSV1716 (MOI 5) treatment. The dotted line indicates the point at which significant upregulation can be seen (1.5-fold increase in expression) Data are means +SEMs, n = 3. (**c**) JJN-3 cells were infected with control or HSV1716 MOI 5. After 24 h, the total RNA was extracted and cDNA was synthesized. A PrimePCR^TM^ SYBR^®^ Green-based apoptotic microarray (BIO-RAD) was performed to detect apoptotic gene expression via RT-qPCR.

**Figure 3 viruses-15-00603-f003:**
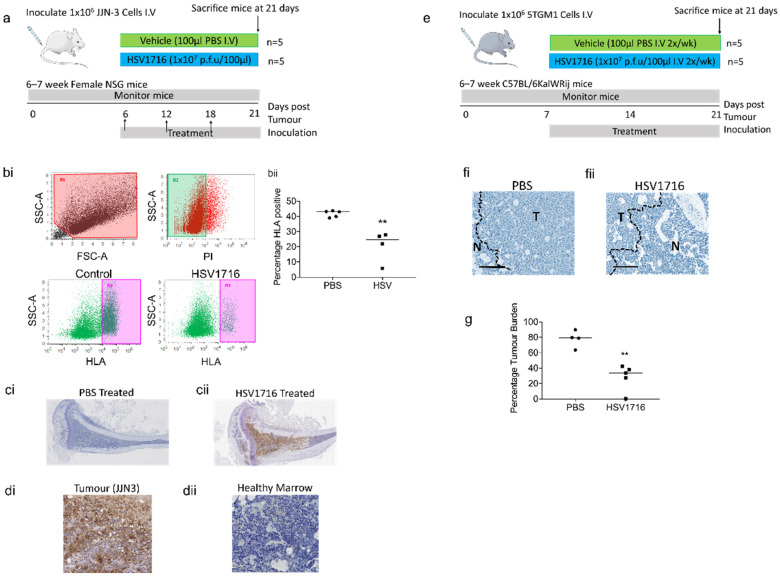
HSV1716 therapy significantly reduces tumor burden in murine models of myeloma. Myeloma xenograft model, (**a**) 6–7-week-old female NSG mice were inoculated with 1 × 10^6^ JJN-3 cells i.v. via the tail vein. Six days post-tumor inoculation, mice were treated with PBS (vehicle control) or HSV1716 1 × 10^7^ p.f.u/100µL i.v. Mice were treated at days 12 and 18, then sacrificed at day 21. (**bi**) At sacrifice, BM samples were flushed from the left tibiae of JJN-3 xenograft mice, labeled with APC-conjugated anti-HLA and analyzed by flow cytometry. Immediately prior to the analysis, the cells were labeled with PI and the viable population was selected (R2). The HLA positivity was then calculated using R3. (**bii**) The average percentage of HLA positivity (tumor burden) was calculated and a significant reduction in tumor burden was seen following the HSV1716 treatment. Data are means, n = 5 mice per group, ** = *p* < 0.01 unpaired T-test with respect to control. (**ci**) At sacrifice, the tibial bones were fixed and decalcified before paraffin sections were stained for HSV viral particles. Positive staining was only seen in HSV1716-treated mice. Composite images of left tibiae from JJN-3 bearing mice treated with (**ci**) PBS or (**cii**) HSV1716, taken at ×2, Bar = 1 mm. In higher magnification images of HSV1716-treated tibiae, viral replication can only be detected in myeloma-infiltrated areas (**di**) and not primarily normal regions of the same bone (**dii**), ×20, bar = 100 µm. In the myeloma syngeneic model, (**e**) 6–7-week-old female C57BL/6KalWRij mice were inoculated with 1 × 10^6^ 5TGM1 cells i.v. via the tail vein. At one week post-tumor inoculation, mice were treated with PBS (vehicle control) or HSV1716 1 × 10^7^ p.f.u/100µL i.v. Mice were treated twice/week and sacrificed after 3 weeks. At sacrifice, tibial bones were fixed and decalcified before paraffin sections were stained with hematoxylin and eosin. The tumor percentage area was assessed using random images and ScanScope software, with composite images of tibiae from 5TGM1-bearing mice treated with PBS (**fi**) or HSV1716 (**fii**) ×10, bar = 200 µM. (**g**) The average percentage of the tumor burden area was calculated and a significant reduction in tumor burden was seen following HSV1716 treatment. Data are means, n = 5 mice per group, ** = *p* < 0.01 unpaired T-test with respect to control.

## Data Availability

All relevant data can be requested by contacting the corresponding author.

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
