# Peer review of "HSV1716 Prevents Myeloma Cell Regrowth When Combined with Bortezomib In Vitro and Significantly Reduces Systemic Tumor Growth in Mouse Models"

_viruses, 2023, doi:10.3390/v15030603_

Round 1
Reviewer 1 Report (Previous Reviewer 2)
Review of resubmitted version of manuscript “HSV1716 Prevents Myeloma Cell Regrowth when Combined with Bortezomib In Vitro and Singly Reduces Systemic Tumour Growth in Mouse Models“ by Simon Tazzyman et al..
In the resubmitted version of the manuscript the authors have addressed the concerns raised in my review of the original manuscript in an adequate fashion. The manuscript has improved significantly and is of interest especially with regard to the two murine myeloma models showing reduction in tumor burden after treatment with HSV1716.
Author Response
Rebuttal for viruses-2218930
Reviewer 1
Minor comments
Are the conclusions supported by the results? (can be improved)
(x) English language and style are fine/minor spell check required
We have now slightly modified the conclusions and spell checked the manuscript.
The study primarily evaluates effects of HSV1716 in oncolytic therapy against myeloma. The data demonstrate infection of various myeloma cell lines and induction of cell death. Data suggest, but do not directly demonstrate, apoptosis is the mechanism of death. In vivo models and combined treatment in vitro with bortezomib also were tested. For the most part, the data are sound and appropriately interpreted such that the results presented support the conclusions reached. There are a couple of minor issues to address.
- The presence of ICP0 and ICP8 transcripts demonstrates that viral transcription occurs in the myeloma cells, but not that viral replication has occurred. This would require evidence that viral titers have increased after infecting the cells. This interpretation should either be modified to appropriately reflect the results, or additional experiments quantifying viral replication should be performed.
We thank the reviewer for the second review. We have now modified the text in several places to reflect this interpretation.
- Fix y-axis scale for Fig. 1ei.
We thank the reviewer for pointing out this error and we have now corrected it.
- Typo - "HSV176" on line 289.
We thank the reviewer for pointing out this error and we have now corrected it.
- Panels and text in Figure 3 is illegibly small. Please increase sizes for all panels and fonts.
We have now increased text font size for all panels in Figure 3.

Reviewer 2 Report (Previous Reviewer 1)
The study primarily evaluates effects of HSV1716 in oncolytic therapy against myeloma. The data demonstrate infection of various myeloma cell lines and induction of cell death. Data suggest, but do not directly demonstrate, apoptosis is the mechanism of death. In vivo models and combined treatment in vitro with bortezomib also were tested. For the most part, the data are sound and appropriately interpreted such that the results presented support the conclusions reached. There are a couple of minor issues to address.
1. The presence of ICP0 and ICP8 transcripts demonstrates that viral transcription occurs in the myeloma cells, but not that viral replication has occurred. This would require evidence that viral titers have increased after infecting the cells. This interpretation should either be modified to appropriately reflect the results, or additional experiments quantifying viral replication should be performed.
2. Fix y-axis scale for Fig. 1ei.
3. Typo - "HSV176" on line 289.
4. Panels and text in Figure 3 is illegibly small. Please increase sizes for all panels and fonts.
Author Response
Rebuttal for viruses-2218930
Reviewer 2
Minor comments
(x) English language and style are fine/minor spell check required
We have now spell checked the manuscript.
Review of resubmitted version of manuscript “HSV1716 Prevents Myeloma Cell Regrowth when Combined with Bortezomib In Vitro and Singly Reduces Systemic Tumour Growth in Mouse Models“ by Simon Tazzyman et al..
In the resubmitted version of the manuscript the authors have addressed the concerns raised in my review of the original manuscript in an adequate fashion. The manuscript has improved significantly and is of interest especially with regard to the two murine myeloma models showing reduction in tumor burden after treatment with HSV1716.
We thank the reviewer for the second review.

This manuscript is a resubmission of an earlier submission. The following is a list of the peer review reports and author responses from that submission.
Round 1
Reviewer 1 Report
This manuscript describes experiments evaluating the impact of HSV1716 oncolytic viral therapy on myeloma cells in vitro and in xenograft models. Experiments show that HSV1716 induces cell death of myeloma cell lines and primary patient cells while not impacting viability of "normal" cells. The death correlates with Annexin V and/or PI positive staining in flow cytometry and an increase in FasL transcription. Administration of HSV1716 to mice xenografted with either human or murine myeloma cell lines correlated with a reduction in tumor burden and a possible improvement in bone density. A single additional experiment testing combined HSV1716 and bortezomib treatment on tumor cells in vitro was shown.
Overall, the experiments shown could increase knowledge of HSV1716 oncolytic viral therapy for myeloma. The findings would support previous publications in this area. However, there are a number of overstated interpretations and conclusions. Additionally, a single experiment is insufficient to support highlighting the proposed combination therapy. If this is the intended major conclusion of the manuscript (it is in the title), then additional experiments are necessary - especially analysis of therapeutic potential in xenograft models. My concerns, both major and minor, are listed below:
Major:
1. There is no in vivo analysis of combined bortezomib and HSV1716 therapy. Since this is in the title of the manuscript, and apparently a major conclusion, this seems like a glaring omission.
2. There is insufficient data provided (based on Annexin V/PI and FasL RT-PCR) to support the conclusion that infected myeloma cells die by apoptosis. Additional experiments are needed.
3. Gene expression analysis is not sufficient to conclude that viral replication is occurring. The RT-PCR analyses should be paired with an analysis of viral genome replication and/or plaque assay on a permissive cell line.
Minor:
1. The title has a typo – Mu-Rine; another – lymphocvtes (line 36). There are several, but most are relatively minor.
2. One assumes that the novelty here is combination therapy, so this should be highlighted in the abstract.
3. Please provide a brief description of the experiments in the figure legend for 1C. Also add a detailed description of the experiment to the methods section.
4. This statement on lines 180-181 is vague and unclear and needs to be re-written: All samples treated with virus had higher levels than the same patient cells in control treated group.
5. Labels on Fig. 3b.i are difficult to read and should have the font size increased.
Reviewer 2 Report
Review of Manuscript “HSV1716 When Combined with Bortezomib Prevents Myeloma Cell Regrowth and Significantly Reduces Tumour Burden in Mu-Rine Models of Myeloma “ by Simon Tazzyman et al..
The present manuscript addresses the effects of modified oncolytic herpes simplex virus HSV1716 (SEPREHVIR®) on myeloma cell lines in vitro and in both a xenograft model and a syngeneic murine model of myeloma in vivo. In vitro infection of myeloma cell lines with HSV1716 resulted in pronounced cell death most probably caused by apotosis. Interestingly, an increase in cell death was also observed in primary myeloma or plasma cell leukemia, CD138+, patient cells, but not in CD138+ cells from healthy donors. The two murine myeloma models showed reduction in tumor burden after treatment with HSV1716.
As the authors concede, other recent publications have already demonstrated the potential of oncolytic HSV for the treatment of myelomas. Thus the findings of the paper are not really novel, but may be taken more as a support of these other recent findings adding some details.
In general, the paper is well written and understandable, but the overall structure may be improved with the experiments performed described in a more detailed fashion.
In addition, the major and minor points listed below should be addressed in a revised version of the manuscript.
Major points:
Several of the figures in the paper have flaws or are of rather low quality:
Fig. 1 c(I) and c(II): no units on abscissa and only rudimentary explanation in figure legend.
Fig. 1 g: is where a special reason that the percentage of PI negative cells is shown in this figure, whereas the number of PI positive cells is shown in all other figures?
Fig. 2 a(i): the resolution of this figure is too low.
Fig. 2 a(ii): the labeling of the bars is not correct. Bars “Annexin V“ for example should read “Annexin V positive, PI negative“, since “Annexin V“ only would normally refer to the total percentage of Annexin V positive cells.
Fig. 3 b(i): the resolution of this figure is much too low, not readable.
Line 261: should probably read Figures 3.d.i, 3.d.
Lines 321 to 325: it is very difficult to follow the argumentation line here. The terminology “live“ and “dead“ virus is rather unusual among virologists and should be specified in detail. If a replication deficient virus was meant with “dead“ virus, when the cytotoxicity would rather be dependent on at least initial HSV replication in contrast to what is stated in the text.
Discussion section: A section with the results of the application of HSV1716 compared to those obtained with other kinds of kinds of oncolytic viruses (e.g. adenoviruses, vaccinia virus, reovirus) in the treatment of multiple myeloma should be included in the discussion section.
Minor points:
Line 186: typo, delete “for“.
Line 193: should probably read “increase in PI staining“ instead of “decrease in PI staining“ (PI negative cells are shown in Fig. 1 g).
Line 310: term “HSV viruses“ – the letter V in the abbreviation HSV already stands for “virus“.
Line 316: ICP0 is not a viral replication gene involved in HSV DNA replication but an immediate early transactivator gene.
Lines 321 to 325: number of reference should be included
Reviewer 3 Report
In agreement with previous published data Tazzyman et al are showing that modified herpes simplex virus HSV1716 (SEPREHVIR®) can infect and induce potent cell death in human myeloma cell lines and primary patients.
It is not clear whether the authors are detecting active viral replication in myeloma cells or killing without replication. Plaque assays should be performed to highlight, if present, differences with the previous virus used by Ghose et. al. Mouse cells are normally not susceptible to HSV infection, if this is the case the authors should show active infection in murine myeloma cells (5TGM1) both in vitro and in vivo. If lack of infection/replication is observed the authors should somehow justify the strong anti-myeloma effects they are observing in in vivo animal model, including but not limited to immune system activation. Few sentences in both introduction and conclusions should be rephrased including but not limited to: " the use of oncolytic HSVs have been published in myeloma, however, recent investigations show that oncolytic HSVs can infect myeloma cells, induce apoptosis, induce oncolysis in cell lines and primary patient samples and reduce tumor burden in subcutaneous xenograft models of myeloma"
Many sentences are somehow contradicting mainly when referred to the previous published literature.
In conclusion this is a nice confirmatory manuscript which highlights few interesting differences, but I think the authors should have a deeper investigation of them